# Cumulative effect of metabolic syndrome on the risk of retinal vein occlusion in young patients: A nationwide population-based study

Yeji Kim[1], Chul Gu Kim[1], Jong Woo Kim[1], Kyungdo Han[2]*, Jae Hui Kim[1]*

1 Kim's Eye Hospital, Seoul, South Korea, 2 Department of Statistics and Actuarial Science, Soongsil University, Seoul, Republic of Korea

* hkd917@naver.com (KH); kimoph@gmail.com (JHK)

**Data Availability Statement:** Access to raw data from the Korean Health Insurance Review and Assessment (HIRA) service is regulated by the Rules for Data Exploration and Use of the HIRA.

## Abstract

This study aimed to investigate the impact of the cumulative burden of metabolic syndrome (MetS) on the incidence of retinal vein occlusion (RVO) in young adults. We included 1,408,093 subjects aged ≥20 and <40 years without a history of RVO who underwent four consecutive annual health examinations during 2009–2012 from the database of the Korean National Health Insurance Service. The metabolic burden was evaluated based on the cumulative number of MetS diagnoses at each health examination (0–4 times) and the cumulative number of each MetS component diagnosed at each health examination (0–4 times per MetS component). Cox proportional hazards models were used to estimate the risk of RVO according to metabolic burden. The risk of RVO was positively correlated with the cumulative number of MetS diagnoses over the four health examinations. All five MetS components were independently associated with an increased risk of RVO. Subgroup analysis for the impact of MetS on RVO occurrence revealed that MetS had a greater impact on female subjects (*P* <0.001). Prompt detection of metabolic derangements and their treatment might be important to decrease the risk of RVO in young adults, especially women.

## Introduction

Retinal vein occlusion (RVO) is the second most common retinal vascular disease, with an estimated prevalence of 0.4% in branch retinal vein occlusion (BRVO) and 0.08% in central retinal vein occlusion (CRVO) [1, 2]. The incidence of RVO increases with age, frequently seen in people older than 65 years [3, 4]. Traditional risk factors for RVO include advancing age and systemic conditions such as hypertension, arteriosclerosis, diabetes mellitus, hyperlipidemia, vascular cerebral stroke, blood hyperviscosity, and thrombophilia [5]. Advanced age is regarded as a basic risk factor, and metabolic syndromes (MetS), such as hypertension, diabetes mellitus, and hyperlipidemia, and is known as a strong risk factor for RVO [5]. Recently, a nationwide population-based study demonstrated that MetS and each of its diagnostic criteria increase the risk of RVO occurrence [6].

Data are available from the Health Insurance Review and Assessment Service database for researchers who meet the criteria for access to confidential data after receiving approval from the HIRA Data Access Committee. The HIRA data can be obtained using the website (http://opendata.hira. or.kr).

**Funding:** The author(s) received no specific funding for this work.

**Competing interests:** The authors have declared that no competing interests exist.

As the incidence of RVO is lower in younger adults, information regarding the risk factors of RVO in the younger population is limited. The etiology of RVO in young patients is likely multifactorial. Previous studies have reported that younger adults were more likely to have nontraditional risk factors, such as physical or psychological stress and hematologic abnormalities [7], whereas other studies have reported that atherosclerotic diseases remain the most commonly associated systemic diseases in patients aged ≤50 years [8]. A substantial increase in the prevalence of MetS has been reported in healthy young adults [9]. In addition, a considerable proportion (7.5–26.2%) of CRVO patients were reported to be under 50 years old [7, 10]. Since visual disturbance in young adults may lead to a significant impact on social participation [11], analyzing the association between MetS and the occurrence of RVO in young patients may have clinical importance.

In the present study, we investigated the association between MetS and the risk of RVO development in young Korean patients through a nationwide, large-scale retrospective cohort study based on the Korean National Health Insurance Database.

## Materials and methods

The methods of Ahn et al. were grossly referenced in this study [12]. This study was approved by the Institutional Review Board (IRB) of Kim's Eye Hospital (Kim's Eye Hospital IRB; IRB number KEH 2022-04-008), and was conducted according to the principles of the Declaration of Helsinki. The need for informed consent was waived by the IRB of Kim's Eye Hospital because the study did not include any identifiable information about the subjects. The Korean National Health Insurance Corporation has allowed authors to use the database based on the approved IRB. The data access dates for research purposes were June 2, 2022

### Study design and data sources

The data analyzed in this study were derived from the National Health Information Database (NHID), which includes all data from the NHIS covering the entire population of the Republic of Korea. Briefly, all insured Korean adults, except 3% of the population covered by the Medical Aid program, were recommended to undergo standardized biennial medical examinations consisting of detailed surveys of demographics, medical histories, lifestyle questionnaires, vital signs, anthropometric measurements, and laboratory tests. In addition to the regular health examination records, the Korean NHIS collects sociodemographic data, income-based insurance contributions, prescription records, inpatient and outpatient usage, and the date of death of all insured Koreans in the NHID [13, 14].

A total of 1,571,091 adults aged ≥20 and <40 years who underwent four serial health examinations between January 1, 2009, and December 31, 2012, were selected from the NHID. The database was collected over four consecutive years from the first health examination. Individuals with a history of RVO, missing health examination data or covariates, or with a diagnosis of RVO within one year from the last health examination (index date) were excluded (Fig 1). A one-year lag period was implemented to enhance the causality of the study. This was done to mitigate the possibility of delayed RVO diagnoses due to delayed eye examinations and to exclude individuals who received regular checkups over a four-year period and passed away within one year thereafter. By excluding these possibilities, this study aimed to increase the causality of the research, measure exposure over a four-year period, and observe the occurrence of RVO one year after the measurements were taken. The washout period was defined as the period from 2002 until the last health examination was conducted, encompassing a duration of 10 years or more (from 2002 to 2012–2015).

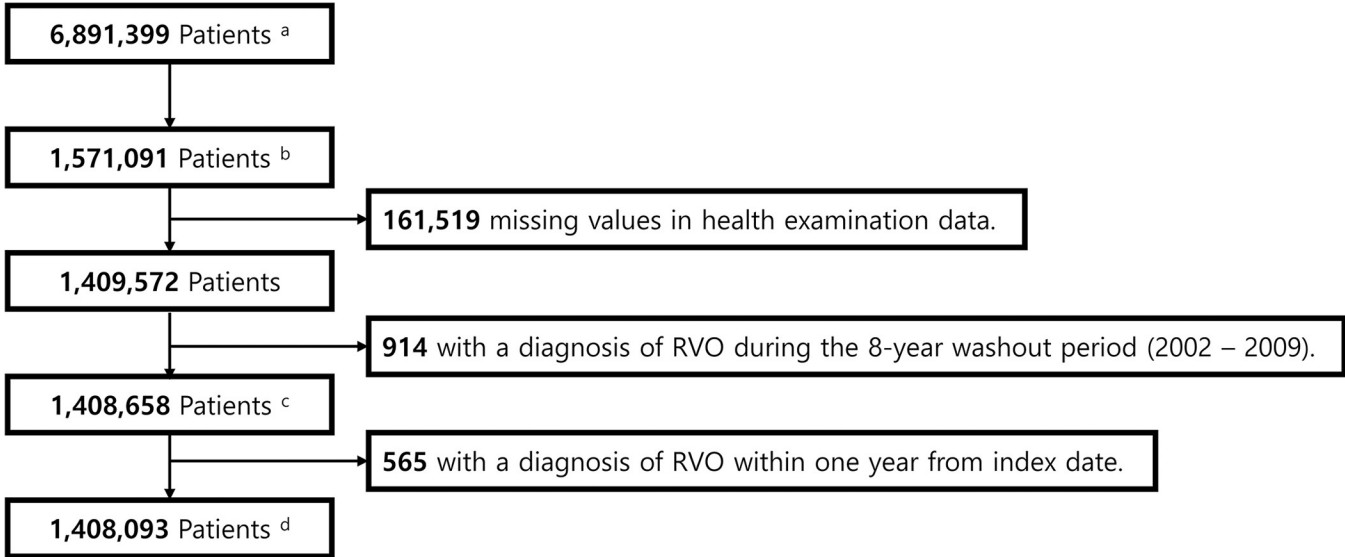

**Fig 1. Flow diagram of the selection of the study population from the National Health Information Database.** a Patients aged ≥20 years and <40 years who participated in the National Health Insurance Service at least once between 2009 and 2012. b Patients who received a health examination between January 1, 2009, and December 31, 2012 and have subsequently undergone four consecutive examinations thereafter (For example, for 2009, participants who underwent annual health examinations continuously for four years from 2009 to 2012). c Patients diagnosed with retinal vein occlusion (KCD code H34.8 corresponding to ICD-10-clinical modification code 362.35, CRVO, or 362.36, venous tributary (branch) occlusion) during the washout period were excluded. d Patients diagnosed with retinal vein occlusion within one year from the index date were excluded (For example, for 2009, individuals who received annual health examinations from 2009 to 2012 and experienced RVO or passed away in 2013 were excluded). *CRVO* central retinal vein occlusion.

In national health examinations, waist circumference, blood pressure, and glucose and cholesterol levels are measured annually, with very few cases of missing data. Individuals with missing data were excluded from the analysis, and only those participants who had complete data for all MetS components over a four-year period were enrolled in the study. Ultimately, 1,408,093 adults were included in the analysis.

### Evaluation of metabolic syndrome and the influence of cumulative metabolic burden

The MetS was defined using the modified waist circumference criteria of the Korean Society for the Study of Obesity and the guidelines of the National Cholesterol Education Program Third Adult Treatment Panel (NCEP-ATP III) as the presence of ≥3 of the following: increased WC [≥90 cm in males or ≥85 cm in females], elevated TG [≥150 mg/dL (1.7 mmol/L) or drug treatment for elevated TG], low HDL-C [<40 mg/dL (1 mmol/L) in males and <50 mg/dL (1.3 mmol/L) in females or drug treatment for low HDL-C], elevated blood pressure [systolic blood pressure ≥130 mmHg or diastolic blood pressure ≥85 mmHg or current use of antihypertensives], and impaired fasting glucose [fasting plasma glucose ≥100 mg/dL (5.6 mmol/L) or current use of anti-diabetes] [15–17].

At each health examination, the presence of MetS and the number of fulfilled MetS components were calculated. "Metabolic burden" was defined in the following ways during four health examinations: (1) cumulative number of MetS diagnosed at each health examination (0–4 times); and (2) cumulative number of each MetS component diagnosed at each health examination (0–4 times per MetS component).

### Covariates, follow-up, and clinical outcomes

The baseline characteristics of the individuals were designated as the data of the last health examination (index date), which comprised sociodemographic data, income-based insurance contributions, laboratory results, anthropometric measurements, comorbidities (diabetes mellitus, hypertension, dyslipidemia, and chronic kidney disease), and answers to lifestyle questionnaires. We investigated the risk of new-onset RVO using the Korean Standard Classification of Diseases Version 6 (KCD-6; an adapted version of the International Classification of Disease, Tenth Revision [ICD-10] codes) and inpatient and outpatient records. Chronic kidney disease was defined as an estimated glomerular filtration rate (eGFR) < 60 mL/min/1.73m$^2$. The follow-up period was the time from the index date to the occurrence of RVO or December 31, 2018, whichever occurred first.

### Statistical analysis

Data are summarized as mean ± standard deviation for continuous variables and number (%) for categorical variables. One-way analysis of variance and the chi-square test were used to evaluate significant differences in baseline characteristics among groups categorized by the number of MetS components. The incidence rate of RVO was computed by dividing new-onset RVO cases by the total follow-up duration and was presented as per 1000 person-years (PY). The association between MetS status frequency and RVO incidence was estimated using Cox proportional hazards regression models. The risk of RVO according to the number of patients with MetS compared with the non-MetS group was expressed as HRs with 95% CIs. Model 1 represented an unadjusted risk, and Model 2 was adjusted for age and sex. Model 3 was adjusted for age; sex; smoking status (never smoker, ex-smoker, or current smoker); alcohol intake (non-, mild, or heavy drinker, g/day); regular exercise (performing >30 min of moderate physical activity ≥5 times a week or >20 min of vigorous physical activity ≥3 times a week); and low-income level (income in the lower 20% of the entire Korean population of subjects supported by the medical aid program) [18]. Model 4 was additionally adjusted for systolic blood pressure, fasting glucose, the logarithm of TG, and HDL-C levels. A *P* value of less than 0.05 was considered statistically significant. Data collection and statistical analysis was performed using SAS version 9.4 (SAS Institute, Cary, NC).

## Results

Table 1 presents the baseline characteristics of 1,408,093 individuals categorized according to the number of MetS diagnoses over four health examinations. The mean age of the total study population was 32.45 ± 4.11 years, and 1,011,343 (71.8%) were men. The study participants were classified into five groups according to the number of MetS diagnostic criteria met during four consecutive annual health examinations. The number of individuals included in each group was 1,072,217 (76.1%), 165,760 (11.8%), 80,501 (5.7%), 50,712 (3.6%), and 38,903 (2.8%) (0, 1, 2, 3, and 4 times, respectively).

### Accumulation of metabolic burden and the risk of RVO

Table 2 shows the incidence of RVO according to the number of MetS and the number of MetS components during the four health examinations. The cumulative number of MetS diagnoses at each health examination and the risk of RVO showed a positive correlation: adjusted HRs with 95% CIs of groups meeting the diagnostic criteria of MetS 1, 2, 3, and 4 times compared to 0 times were 1.20 (1.03–1.39), 1.35 (1.12–1.63), 1.58 (1.28–1.95), and 1.71 (1.36–2.15), *P* for trend <0.001. Fig 2A shows the relation.

**Table 1. Baseline characteristics of the total study population according to the cumulative number of MetS diagnosed at each health examination (0 to 4 times).**

| | Total | The number of the presence of the metabolic syndrome | | | | | p |
|---|---|---|---|---|---|---|---|
| | | 0 | 1 | 2 | 3 | 4 | |
| No. of participants (%) | 1,408,093 (100.0) | 1,072,217 (76.1) | 165,760 (11.8) | 80,501 (5.7) | 50,712 (3.6) | 38,903 (2.8) | |
| Age (years) | 32.45 ± 4.11 | 32.02 ± 4.16 | 33.5 ± 3.71 | 33.91 ± 3.55 | 34.2 ± 3.47 | 34.63 ± 3.35 | < .0001 |
| Sex | | | | | | | < .0001 |
| Male | 1,011,343 (71.8) | 703,191 (65.6) | 149,321 (90.1) | 74,779 (92.9) | 47,505 (93.7) | 36,547 (93.9) | |
| Female | 396,750 (28.2) | 369,026 (34.4) | 16,439 (9.9) | 5,722 (7.1) | 3,207 (6.3) | 2,356 (6.1) | |
| Smoking | | | | | | | < .0001 |
| Never smoker | 672,683 (47.8) | 573,880 (53.5) | 53,442 (32.2) | 22,564 (28.0) | 13,197 (26.0) | 9,600 (24.7) | |
| Ex-smoker | 206,809 (14.7) | 145,501 (13.6) | 29,763 (18.0) | 15,238 (18.9) | 9,396 (18.5) | 6,911 (17.8) | |
| Current smoker | 528,601 (37.5) | 352,836 (32.9) | 82,555 (49.8) | 42,699 (53.0) | 28,119 (55.5) | 22,392 (57.6) | |
| Alcohol consumption[a] | | | | | | | < .0001 |
| Non-drinker | 444,733 (31.6) | 362,414 (33.8) | 41,844 (25.2) | 19,140 (23.8) | 11,887 (23.4) | 9,448 (24.3) | |
| Mild to moderate drinker | 832,667 (59.1) | 627,253 (58.5) | 101,847 (61.4) | 49,433 (61.4) | 30,905 (60.9) | 23,229 (59.7) | |
| Heavy drinker | 130,693 (9.3) | 82,550 (7.7) | 22,069 (13.3) | 11,928 (14.8) | 7,920 (15.6) | 6,226 (16.0) | |
| Regular exercise[b] | 248,553 (17.7) | 187,096 (17.5) | 30,730 (18.5) | 14,802 (18.4) | 9,178 (18.1) | 6,747(17.3) | < .0001 |
| Low income[c] | 72 (0.0) | 54 (0.0) | 7 (0.0) | 5 (0.0) | 4 (0.0) | 2 (0.0) | < .0001 |
| Comorbidities | | | | | | | |
| Diabetes mellitus | 29,670 (2.1) | 7,387 (0.7) | 4,494 (2.7) | 4,138 (5.1) | 4,847 (9.6) | 8,804 (22.6) | < .0001 |
| Hypertension | 107,787 (7.7) | 38,336 (3.6) | 21,002 (12.7) | 16,247 (20.2) | 14,642 (28.9) | 17,560(45.1) | < .0001 |
| Dyslipidemia | 125,484 (8.9) | 59,578 (5.6) | 23,902 (14.4) | 15,779 (19.6) | 12,509 (24.7) | 13,716(35.3) | < .0001 |
| CKD | 4,993 (0.4) | 3,350 (0.3) | 604 (0.4) | 370 (0.5) | 272 (0.5) | 397(1.0) | < .0001 |
| Laboratory findings | | | | | | | |
| BMI (kg/m2) | 23.58 ± 3.65 | 22.53 ± 2.97 | 25.71 ± 3.16 | 27.22 ± 3.31 | 28.42 ± 3.45 | 29.75 ± 3.68 | < .0001 |
| Waist circumference (cm) | 79.61 ± 9.99 | 76.77 ± 8.55 | 85.72 ± 7.97 | 89.37 ± 8.06 | 92.19 ± 8.24 | 95.31 ± 8.51 | < .0001 |
| Systolic BP (mmHg) | 119.07 ± 12.78 | 116.46 ± 11.69 | 124.90 ± 11.80 | 127.95 ± 12.04 | 130.32 ± 12.56 | 133.23 ± 13.64 | < .0001 |
| Diastolic BP (mmHg) | 74.93 ± 9.19 | 73.19 ± 8.43 | 78.66 ± 8.65 | 80.84 ± 8.99 | 82.54 ± 9.56 | 84.72 ± 10.38 | < .0001 |
| Fasting glucose (mg/dL) | 92.00 ± 16.06 | 89.46 ± 11.03 | 95.81 ± 16.30 | 99.21 ± 21.09 | 104.06 ± 28.76 | 115.26 ± 42.11 | < .0001 |
| Total cholesterol (mg/dL) | 189.34 ± 34.10 | 184.93 ± 31.99 | 199.84 ± 35.40 | 205.12 ± 36.44 | 208.17 ± 37.49 | 208.97 ± 39.85 | < .0001 |
| HDL cholesterol (mg/dL) | 56.08 ± 18.47 | 58.79 ± 18.44 | 49.46 ± 16.22 | 46.71 ± 15.14 | 44.96 ± 14.05 | 43.33 ± 15.04 | < .0001 |
| LDL cholesterol (mg/dL) | 108.86 ± 32.47 | 106.03 ± 30.48 | 116.87 ± 34.93 | 119.19 ± 36.61 | 119.73 ± 38.93 | 117.09 ± 40.87 | < .0001 |
| eGFR (ml/min/1.73m2) | 101.68 ± 68.64 | 102.54 ± 69.76 | 99.46 ± 67.04 | 98.73 ± 65.50 | 97.97 ± 58.14 | 98.45 ± 62.49 | < .0001 |
| *Triglyceride (mg/dL) | 105.81 | 89.77 | 155.03 | 186.14 | 211.29 | 243.67 | < .0001 |
| | (105.70–105.92) | (89.68–89.86) | (154.64–155.43) | (185.48–186.80) | (210.37–212.21) | (242.47–244.88) | |

Data are presented as mean ± SD or No. (Percentages)

Percentages may not total 100 because of rounding

MetS metabolic syndrome, CKD chronic kidney disease, BMI body mass index, BP blood pressure, HDL high-density lipoprotein, LDL low-density lipoprotein, eGFR estimated glomerular filtration rate

[a] Alcohol consumption denotes as following

Non-drinker: alcohol consumption 0 g

Mild to moderate drinker: alcohol consumption >0 g to <30 g per day

Heavy drinker: alcohol consumption ≥30 g per day

[b] Regular exercise denotes performing >30 min of moderate-intensity exercise (e.g., brisk pace walking, tennis doubles, or bicycling leisurely) ≥ 5 times a week or >20 min of vigorous-intensity exercise (e.g., running, climbing, fast cycling, or aerobics) ≥ 3 times a week

[c] Low income denotes income of the lower 20% among the entire Korean population of subjects supported by the Medical Aid program

*Geometric mean (95% CI)

**Table 2. The risk of retinal vein occlusion according to the cumulative number of MetS and each component diagnosed during four health examinations (0 to 4 times).**

| The number of meeting the component | No. of participants | RVO | IR (1000PY) | HR (95% CI) | | | |
|---|---|---|---|---|---|---|---|
| | | | | Model1 | Model2 | Model3 | Model4 |
| **Obesity (BMI ≥25)** | | | | | | | |
| 0 | 871,556 | 834 | 0.20 | 1.00 (Reference) | 1.00 (Reference) | 1.00 (Reference) | 1.00 (Reference) |
| 1 | 89,802 | 94 | 0.22 | 1.10 (0.89–1.36) | 1.06 (0.85–1.31) | 1.05 (0.85–1.31) | 1.02 (0.82–1.26) |
| 2 | 70,160 | 90 | 0.27 | 1.36 (1.09–1.68) | 1.29 (1.03–1.60) | 1.28 (1.03–1.60) | 1.22 (0.98–1.52) |
| 3 | 79,588 | 85 | 0.23 | 1.13 (0.90–1.41) | 1.05 (0.84–1.32) | 1.05 (0.84–1.32) | 0.98 (0.79–1.23) |
| 4 | 296,987 | 490 | 0.35 | 1.72 (1.54–1.93) | 1.55 (1.38–1.75) | 1.56 (1.39–1.75) | 1.33 (1.18–1.50) |
| **Metabolic syndrome** | | | | | | | |
| 0 | 1,072,217 | 1,014 | 0.20 | 1.00 (Reference) | 1.00 (Reference) | 1.00 (Reference) | 1.00 (Reference) |
| 1 | 165,760 | 222 | 0.28 | 1.39 (1.20–1.61) | 1.27 (1.10–1.47) | 1.29 (1.11–1.49) | 1.20 (1.03–1.39) |
| 2 | 80,501 | 133 | 0.35 | 1.71 (1.43–2.05) | 1.53 (1.27–1.83) | 1.55 (1.29–1.87) | 1.35 (1.12–1.63) |
| 3 | 50,712 | 109 | 0.45 | 2.22 (1.83–2.71) | 1.95 (1.59–2.38) | 1.99 (1.63–2.43) | 1.58 (1.28–1.95) |
| 4 | 38,903 | 115 | 0.62 | 3.06 (2.52–3.71) | 2.62 (2.15–3.19) | 2.69 (2.21–3.27) | 1.71 (1.36–2.15) |
| **Increased waist circumference** | | | | | | | |
| 0 | 1,067,469 | 1,062 | 0.21 | 1.00 (Reference) | 1.00 (Reference) | 1.00 (Reference) | 1.00 (Reference) |
| 1 | 121,985 | 164 | 0.28 | 1.36 (1.15–1.60) | 1.29 (1.10–1.53) | 1.30 (1.10–1.53) | 1.23 (1.04–1.45) |
| 2 | 69,078 | 88 | 0.27 | 1.30 (1.04–1.61) | 1.21 (0.97–1.51) | 1.21 (0.98–1.51) | 1.11 (0.89–1.39) |
| 3 | 59,932 | 93 | 0.33 | 1.58 (1.28–1.96) | 1.47 (1.18–1.81) | 1.47 (1.19–1.82) | 1.29 (1.04–1.60) |
| 4 | 89,629 | 186 | 0.44 | 2.12 (1.82–2.48) | 1.93 (1.65–2.27) | 1.96 (1.67–2.29) | 1.55 (1.31–1.83) |
| **Low HDL-C** | | | | | | | |
| 0 | 1,001,902 | 1,050 | 0.22 | 1.00 (Reference) | 1.00 (Reference) | 1.00 (Reference) | 1.00 (Reference) |
| 1 | 202,652 | 249 | 0.26 | 1.15 (1.01–1.32) | 1.13 (0.98–1.29) | 1.13 (0.99–1.30) | 1.08 (0.94–1.24) |
| 2 | 97,195 | 135 | 0.29 | 1.30 (1.09–1.55) | 1.25 (1.04–1.49) | 1.26 (1.05–1.50) | 1.15 (0.96–1.38) |
| 3 | 61,691 | 77 | 0.26 | 1.16 (0.92–1.47) | 1.10 (0.87–1.38) | 1.11 (0.88–1.40) | 0.99 (0.78–1.25) |
| 4 | 44,653 | 82 | 0.38 | 1.71 (1.36–2.14) | 1.58 (1.26–1.98) | 1.60 (1.27–2.00) | 1.32 (1.05–1.66) |
| **Elevated blood pressure** | | | | | | | |
| 0 | 632,898 | 579 | 0.20 | 1.00 (Reference) | 1.00 (Reference) | 1.00 (Reference) | 1.00 (Reference) |
| 1 | 309,082 | 329 | 0.22 | 1.15 (1.00–1.31) | 1.08 (0.94–1.25) | 1.09 (0.94–1.25) | 1.06 (0.92–1.22) |
| 2 | 204,139 | 241 | 0.25 | 1.26 (1.09–1.47) | 1.16 (0.99–1.36) | 1.17 (1.00–1.37) | 1.10 (0.94–1.29) |
| 3 | 145,957 | 183 | 0.26 | 1.33 (1.13–1.58) | 1.20 (1.01–1.43) | 1.21 (1.02–1.44) | 1.06 (0.88–1.27) |
| 4 | 116,017 | 261 | 0.47 | 2.37 (2.05–2.75) | 2.06 (1.76–2.40) | 2.07 (1.77–2.42) | 1.52 (1.26–1.84) |
| **Impaired fasting glucose** | | | | | | | |
| 0 | 814,304 | 801 | 0.21 | 1.00 (Reference) | 1.00 (Reference) | 1.00 (Reference) | 1.00 (Reference) |
| 1 | 336,845 | 351 | 0.22 | 1.05 (0.92–1.19) | 0.98 (0.87–1.11) | 0.99 (0.87–1.12) | 0.96 (0.85–1.09) |
| 2 | 147,913 | 192 | 0.27 | 1.30 (1.11–1.52) | 1.16 (0.99–1.36) | 1.17 (1.00–1.37) | 1.09 (0.93–1.28) |
| 3 | 68,920 | 109 | 0.33 | 1.57 (1.29–1.92) | 1.35 (1.10–1.66) | 1.36 (1.11–1.67) | 1.18 (0.96–1.45) |
| 4 | 40,111 | 140 | 0.73 | 3.47 (2.90–4.15) | 2.84 (2.37–3.41) | 2.86 (2.38–3.44) | 1.84 (1.46–2.32) |
| **Elevated TG** | | | | | | | |
| 0 | 750,472 | 713 | 0.20 | 1.00 (Reference) | 1.00 (Reference) | 1.00 (Reference) | 1.00 (Reference) |
| 1 | 245,822 | 268 | 0.23 | 1.12 (0.98–1.29) | 1.04 (0.90–1.20) | 1.05 (0.91–1.21) | 1.01 (0.88–1.17) |
| 2 | 152,002 | 188 | 0.26 | 1.26 (1.08–1.48) | 1.11 (0.94–1.31) | 1.13 (0.96–1.34) | 1.05 (0.89–1.25) |
| 3 | 123,657 | 186 | 0.31 | 1.53 (1.30–1.79) | 1.30 (1 .10–1.54) | 1.34 (1.13–1.58) | 1.18 (1.00–1.41) |

(*Continued*)

**Table 2.** (Continued)

| The number of meeting the component | No. of participants | RVO | IR (1000PY) | HR (95% CI) | | | |
|---|---|---|---|---|---|---|---|
| | | | | Model1 | Model2 | Model3 | Model4 |
| 4 | 136,140 | 238 | 0.36 | 1.75 (1.51–2.03) | 1.44 (1.23–1.68) | 1.49 (1.28–1.75) | 1.19 (1.00–1.40) |

Model 1 is unadjusted

Model 2 is adjusted for age, sex

Model 3 is adjusted for age, sex, smoking status, alcohol intake, and regular exercise

Model 4 is adjusted for age, sex, smoking status, alcohol intake, regular exercise, systolic blood pressure, fasting glucose, logarithm of TG, and HDL-C level

RVO retinal vein occlusion, IR incidence rate, PY person-years, HR hazard ratio, CI confidence interval, HDL-C high-density lipoprotein cholesterol, TG triglycerides

All of the five MetS criteria were independently related to the risk of RVO development. With the increase in the number of each MetS diagnostic criterion satisfied, the incidence and risk of RVO also increased (Table 2 and Fig 1B–1F). The impact of each MetS component on RVO occurrence differed. The adjusted HR (95% CI) of each MetS component, when diagnosed four times serially, was as follows: 1.84 (1.46–2.32) for impaired fasting glucose, 1.55 (1.31–1.83) for increased waist circumference (WC), 1.52 (1.26–1.84) for elevated blood pressure, 1.32 (1.05–1.66) for low HDL-C, and 1.19 (1.00–1.40) for elevated TG.

## Subgroup analysis

Additional subgroup analyses for RVO development and the cumulative number of MetS were performed and are presented in Table 3. MetS increased the risk of RVO development greater in the female subgroup than in the male subgroup (adjusted HR (95% CI), 4.76 (2.48–9.15) for

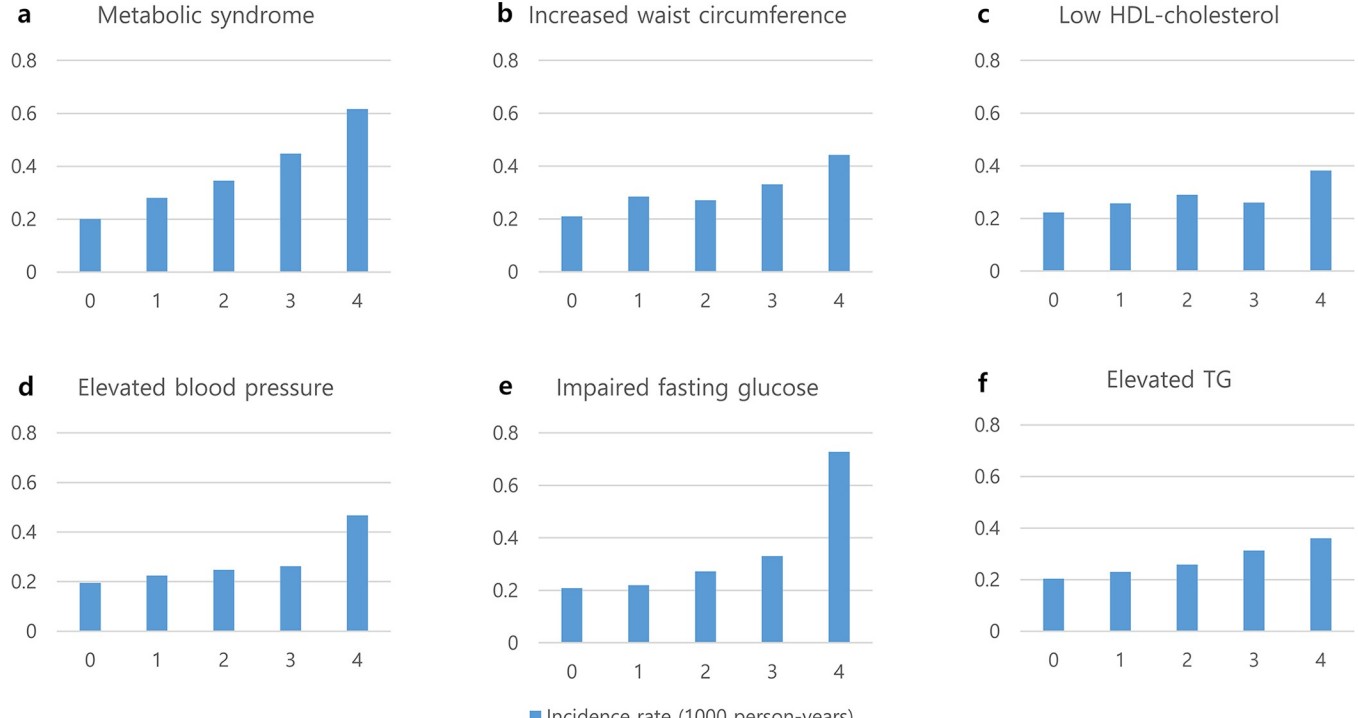

**Fig 2.** Incidence rate of retinal vein occlusion according to the a. cumulative number of MetS diagnosed at each health examination, and b-f. cumulative number of each component of MetS diagnosed at each health examination. *HDL* high-density lipoprotein, *TG* triglycerides.

**Table 3. The risk of retinal vein occlusion according to the cumulative number of MetS in various subgroups.**

| | The number of the presence of the metabolic syndrome | No. of participants | RVO | IR (1000PY) | HR (95% CI) | Interaction P |
|---|---|---|---|---|---|---|
| Age | | | | | | |
| <30 | 0 | 313,144 | 199 | 0.15 | 1.00 (Reference) | 0.1286 |
| | 1 | 24,970 | 9 | 0.08 | 0.61 (0.31–1.20) | |
| | 2 | 9,392 | 11 | 0.28 | 1.86 (0.98–3.52) | |
| | 3 | 5,030 | 5 | 0.24 | 1.44 (0.57–3.66) | |
| | 4 | 3,004 | 5 | 0.39 | 1.90 (0.69–5.18) | |
| ≥30 | 0 | 759,073 | 815 | 0.22 | 1.00 (Reference) | |
| | 1 | 140,790 | 213 | 0.31 | 1.25 (1.07–1.46) | |
| | 2 | 71,109 | 122 | 0.35 | 1.31 (1.08–1.60) | |
| | 3 | 45,682 | 104 | 0.47 | 1.58 (1.27–1.96) | |
| | 4 | 35,899 | 110 | 0.63 | 1.69 (1.33–2.14) | |
| Sex | | | | | | |
| Male | 0 | 703,191 | 727 | 0.22 | 1.00 (Reference) | < .0001 |
| | 1 | 149,321 | 199 | 0.28 | 1.14 (0.97–1.33) | |
| | 2 | 74,779 | 117 | 0.33 | 1.22 (1.00–1.49) | |
| | 3 | 47,505 | 104 | 0.46 | 1.54 (1.24–1.91) | |
| | 4 | 36,547 | 98 | 0.56 | 1.49 (1.17–1.91) | |
| Female | 0 | 36,9026 | 287 | 0.17 | 1.00 (Reference) | |
| | 1 | 16,439 | 23 | 0.30 | 1.61 (1.04–2.48) | |
| | 2 | 5,722 | 16 | 0.60 | 2.94 (1.74–4.97) | |
| | 3 | 3,207 | 5 | 0.33 | 1.42 (0.57–3.57) | |
| | 4 | 2,356 | 17 | 1.56 | 4.76 (2.48–9.15) | |
| Obesity | | | | | | |
| No | 0 | 857,392 | 789 | 0.19 | 1.00 (Reference) | 0.3659 |
| | 1 | 68,993 | 81 | 0.24 | 1.03 (0.81–1.30) | |
| | 2 | 19,324 | 40 | 0.42 | 1.56 (1.12–2.17) | |
| | 3 | 7,126 | 18 | 0.51 | 1.58 (0.97–2.58) | |
| | 4 | 2,873 | 9 | 0.63 | 1.37 (0.68–2.77) | |
| Yes | 0 | 214,825 | 225 | 0.22 | 1.00 (Reference) | |
| | 1 | 96,767 | 141 | 0.31 | 1.26 (1.02–1.56) | |
| | 2 | 61,177 | 93 | 0.32 | 1.21 (0.94–1.55) | |
| | 3 | 43,586 | 91 | 0.44 | 1.49 (1.16–1.93) | |
| | 4 | 36,030 | 106 | 0.62 | 1.68 (1.28–2.20) | |
| Current smoker | | | | | | |
| No | 0 | 719,381 | 704 | 0.21 | 1.00 (Reference) | 0.4681 |
| | 1 | 83,205 | 127 | 0.32 | 1.28 (1.05–1.55) | |
| | 2 | 37,802 | 68 | 0.38 | 1.37 (1.06–1.78) | |
| | 3 | 22,593 | 47 | 0.44 | 1.44 (1.05–1.97) | |
| | 4 | 16,511 | 51 | 0.65 | 1.72 (1.23–2.41) | |
| Yes | 0 | 352,836 | 310 | 0.18 | 1.00 (Reference) | |
| | 1 | 82,555 | 95 | 0.24 | 1.12 (0.89–1.41) | |
| | 2 | 42,699 | 65 | 0.32 | 1.34 (1.02–1.76) | |
| | 3 | 28,119 | 62 | 0.46 | 1.70 (1.27–2.27) | |
| | 4 | 22,392 | 64 | 0.59 | 1.66 (1.21–2.29) | |
| Heavy drink[a] | | | | | | |
| No | 0 | 989,667 | 940 | 0.20 | 1.00 (Reference) | 0.513 |
| | 1 | 143,691 | 192 | 0.28 | 1.18 (1.01–1.39) | |

*(Continued)*

**Table 3.** (Continued)

| | The number of the presence of the metabolic syndrome | No. of participants | RVO | IR (1000PY) | HR (95% CI) | Interaction P |
|---|---|---|---|---|---|---|
| | 2 | 68,573 | 109 | 0.33 | 1.29 (1.05–1.58) | |
| | 3 | 42,792 | 97 | 0.47 | 1.66 (1.32–2.07) | |
| | 4 | 32,677 | 99 | 0.63 | 1.75 (1.37–2.24) | |
| Yes | 0 | 82,550 | 74 | 0.19 | 1.00 (Reference) | |
| | 1 | 22,069 | 30 | 0.28 | 1.32 (0.86–2.03) | |
| | 2 | 11,928 | 24 | 0.42 | 1.78 (1.10–2.87) | |
| | 3 | 7,920 | 12 | 0.32 | 1.18 (0.62–2.25) | |
| | 4 | 6,226 | 16 | 0.53 | 1.55 (0.82–2.90) | |
| Regular exercise[b] | | | | | | |
| No | 0 | 885,121 | 832 | 0.20 | 1.00 (Reference) | 0.2396 |
| | 1 | 135,030 | 183 | 0.28 | 1.21 (1.03–1.43) | |
| | 2 | 65,699 | 98 | 0.31 | 1.21 (0.98–1.51) | |
| | 3 | 41,534 | 86 | 0.43 | 1.50 (1.18–1.90) | |
| | 4 | 32,156 | 97 | 0.63 | 1.70 (1.32–2.18) | |
| Yes | 0 | 187,096 | 182 | 0.21 | 1.00 (Reference) | |
| | 1 | 30,730 | 39 | 0.27 | 1.15 (0.80–1.63) | |
| | 2 | 14,802 | 35 | 0.49 | 1.99 (1.36–2.90) | |
| | 3 | 9,178 | 23 | 0.53 | 1.95 (1.23–3.09) | |
| | 4 | 6,747 | 18 | 0.56 | 1.72 (0.99–3.02) | |

HRs with 95% CI were calculated using a Cox proportional hazards model adjusting age, sex, smoking status, alcohol intake, and regular exercisde.

[a] Heavy drinkers were defined as those who drank more than 30 g/d.

[b] Regular exercise was defined as strenuous physical activity that was performed for at least 30 minutes at least five times a week.

female, 1.49 (1.17–1.91) for male; *P* for interaction <0.0001; when diagnosed four times consecutively). Other variables including obesity, current smoking, heavy drinking, and regular performance did not show significant differences in the subgroup analyses.

## Discussion

RVO, especially CRVO, is a significant cause of visual impairment in individuals of any age [19, 20]. A nationwide, retrospective study using data from the Korean National Health Claims Database, spanning 2007 to 2011, revealed the RVO incidences per 100,000 person-years [21]. For ages 20–29, 30–39, and 40–49, the rates were 4.03, 10.32, and 32.22, respectively. The rates dramatically increased with age, with incidences of 80.22 for ages 50–59 and 172.90 for ages 60–69. This pattern shows a more than twofold increase in incidence every decade from the 20s to the 70s.

Younger patients with RVO have a variable clinical course [22, 23]. Although young patients with RVO generally have a better visual prognosis [7, 24–28], at least 20% of patients are known to develop poor visual outcomes with severe neovascular complications [10, 29]. Therefore, identifying factors associated with the development of RVO in young subjects would be of great value.

To the best of our knowledge, this is the first large nationwide population-based cohort study to report the cumulative effect of MetS and its components on the risk of RVO in young adults. The key findings of this study can be summarized as follows: (1) the cumulative burden of MetS during the four health examinations had a linear correlation with the risk of RVO, (2) the cumulative burden of each MetS component showed a positive association with the risk of

RVO, (3) among the five metabolic components, impaired fasting glucose showed the greatest increase in RVO risks, and (4) MetS increased the risk of RVO more in females.

The association of MetS and its components with RVO has been reported in several previous studies [6, 30]. While these studies analyzed the number of MetS components at a single point, our study investigated the accumulated effects of MetS identified in four consecutive health examinations. Participants who were diagnosed with MetS at one time had a 20% increased risk of RVO compared to those without MetS, whereas participants who were diagnosed with MetS four times had a 71% higher risk of RVO. Such a positive increase in RVO according to the repeated diagnosis of metabolic components may suggest a dose-response perspective for evaluating the risk of retinal microvascular disease. Given the proportional increase in RVO risk according to the degree of metabolic burden with temporal changes, it is plausible that the risk of RVO would be higher in patients diagnosed with MetS more than four times during their health checkups.

Among the five diagnostic components of MetS, the impact of WC on RVO occurrence was particularly significant when individuals with impaired fasting glucose, prediabetes, or diabetes were excluded. In a previous population-based study of individuals aged 20 years and older, elevated blood pressure was the greatest risk and increased WC was the minimal risk for RVO development, revealing adjusted hazard ratios of 1.610 (95% CI 1.589–1.631) for BP and 1.212 (95% CI 1.197–1.227) for WC [6]. The reason for the discrepancy in the results of the present study may be associated with age. Another population-based study revealed that obesity has different effects on the incidence of RVO in the presence and absence of DM. In individuals with DM, a lower body mass index (BMI) and WC were associated with an increased risk of RVO, while a higher BMI and WC were associated with a lower risk of RVO. In those without DM, the correlation was reversed; a lower WC was associated with a lower risk of RVO and vice versa [31]. Since the prevalence of DM is lower in the younger population, our study of younger individuals showed similar results to the group without DM. As the cumulative number of times included in the WC criterion increased during the four health examination checkups, the risk of RVO occurrence increased linearly. Further studies are required to determine whether there is a definitive causal relationship between WC and MetS.

The sex differences are observed in MetS prevalence and cardiovascular disease (CVD) risk [32]. Studies have shown that the risk for CVD in MetS is greater among females than among males [33, 34]. Each component of MetS also has shown sex differences; dyslipidemia was associated with a greater risk for coronary artery disease, and congestive heart failure was more commonly seen as a consequence of hypertension in females than in males [32]. Similarly, the subgroup analysis for the impact of MetS on RVO occurrence has revealed that MetS has a greater impact on female subjects. Several circumstances specific to females, including pregnancy, polycystic ovary syndrome, and oral contraceptive therapy may play a role in increasing these risks. In addition, it is well known that the prevalence of MetS increases with menopause [35], and the risk of RVO is reported to increase in MetS [6]. Furthermore, a recent study showed that early menopause is associated with an increased risk of RVO [36]. However, in premenopausal women with a relatively low risk of MetS, as in the present study, although the exact mechanism is unclear and requires further investigation, there may be risk factors that can cause MetS other than a decrease in estrogen, which may also affect the occurrence of RVO.

The data employed in this study originates from nationwide health examinations administered to the entire cohort of employed individuals aged 20 and above. These examinations are not individually solicited according to personal healthcare requirements, thereby mitigating the potential for differential screening rates among individuals with suboptimal overall health or higher socioeconomic status.

Our study had several limitations. First, using a claims database focused solely on healthcare utilization to identify patients with RVO may have underrepresented asymptomatic cases. Therefore, our data reflected the incidence of clinically diagnosed RVO, potentially underestimating the true occurrence of RVO. Second, the lack of access to hospital-based medical records to validate RVO occurrences and review clinical data compromised the accuracy of the study data. These limitations include the risk of misclassifying diagnoses and the inability to assess factors such as severity, extent, prognosis, or patient comorbidities. Third, owing to the inherent constraints of studies utilizing claims databases, clinical characteristics and interobserver variability in RVO diagnosis were not standardized. Fourth, male patients comprised over two-thirds of the participants, owing to the higher probability of employees undergoing routine health checkups at their workplaces. This sex imbalance introduces a potential selection bias, limiting the representation of both sexes in the broader population. Furthermore, the selection criteria, emphasizing individuals who underwent health assessments four times over four years, might not fully capture the demographics of individuals in their 20s and 30s within the general population.

The major limitation of the present study is that an analysis utilizing the Charlson Comorbidity Index was not performed. Furthermore, the analysis did not account for the use of contraceptives, which is associated with an increased risk of retinal vascular occlusion [37]. Although RVO in younger patients may be related to systemic diseases, such as rheumatic disorders and leukemia, this study did not evaluate the associations with these specific diseases.

## Conclusions

In conclusion, this large-scale population-based cohort study revealed that the cumulative burden of MetS and its components was positively correlated with the risk of RVO in young adults. Given the association found in this study, prompt detection of metabolic derangements and their treatment might be important for decreasing the risk of RVO in young adults, especially in females.

## Supporting information

**S1 Checklist. STROBE statement—checklist of items that should be included in reports of observational studies.**
(DOCX)

## Acknowledgments

The authors wish to thank the National Health Insurance Service for providing the data regarding the original cohort.

## Author Contributions

**Conceptualization:** Jae Hui Kim.

**Data curation:** Yeji Kim, Chul Gu Kim, Jong Woo Kim, Kyungdo Han, Jae Hui Kim.

**Formal analysis:** Yeji Kim, Chul Gu Kim, Jong Woo Kim, Kyungdo Han, Jae Hui Kim.

**Funding acquisition:** Chul Gu Kim, Jong Woo Kim, Jae Hui Kim.

**Investigation:** Yeji Kim, Chul Gu Kim, Jong Woo Kim, Kyungdo Han, Jae Hui Kim.

**Methodology:** Yeji Kim, Kyungdo Han, Jae Hui Kim.

**Project administration:** Kyungdo Han, Jae Hui Kim.

**Resources:** Kyungdo Han, Jae Hui Kim.

**Software:** Kyungdo Han.

**Supervision:** Kyungdo Han, Jae Hui Kim.

**Validation:** Kyungdo Han, Jae Hui Kim.

**Visualization:** Yeji Kim, Jae Hui Kim.

**Writing – original draft:** Yeji Kim, Chul Gu Kim, Kyungdo Han, Jae Hui Kim.

**Writing – review & editing:** Yeji Kim, Jong Woo Kim, Jae Hui Kim.

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
