## [Decision Letter · Decision Letter 0]

24 Oct 2023

PONE-D-23-20018Cumulative effect of metabolic syndrome on the risk of retinal vein occlusion in young patients: a nationwide population-based studyPLOS ONE

Dear Dr. Kim,

Thank you for submitting your manuscript to PLOS ONE. After careful consideration, we feel that it has merit but does not fully meet PLOS ONE’s publication criteria as it currently stands. Therefore, we invite you to submit a revised version of the manuscript that addresses the points raised during the review process.

We look forward to receiving your revised manuscript.

Kind regards,

Se Woong Kang, Ph.D., M.D.

Academic Editor

PLOS ONE

REVIEWER 1

Kim et al. demonstrated the impact of the longitudinal cumulative burden of METS on RVO in a young population. Given the significant impact of eye diseases in the young population, this research holds considerable value. Moreover, the examination of cumulative burden is an interesting aspect of the study. However, there are several points that need to be addressed:

1. While it is acknowledged that disease burden is higher in the young population, it would be beneficial to mention how the RVO incidence in the 20-40 age group compares to the overall population incidence or the incidence in the elderly. Since there have been similar studies conducted in Korea previously, referencing these studies for comparison would enhance the discussion.

2. On line 82, Figure 2 precedes Figure 1 in the manuscript. It is advisable to reverse the order for clarity.

3. Regarding line 82, please provide a rationale for the selection of a 1-year lag period.

4. Regarding Figure 2, does this study include individuals who underwent health examinations four times between 2009 and 2012? If so, the index data should presumably be in 2012. However, in Figure 2, it mentions the exclusion of cases diagnosed with RVO from 2002 to 2009 and within one year from the index date. Does this mean that individuals diagnosed with RVO in 2010 were not excluded? I assume that individuals diagnosed with RVO before the index date might have been excluded. The flow chart requires further clarification to provide a clearer understanding.

5. On line 167, the conclusion "among the five metabolic components, increased WC had the greatest increase in RVO risks" needs further clarification. What criteria were used to reach this conclusion? Among the components, it appears that impaired fasting glucose has the highest risk when there are four times serial abnormalities. Please elaborate.

6. On line 177, the statement, "it can be inferred that the risk of RVO increases more than four times in people diagnosed with MetS during health examinations," requires additional clarification. It is crucial to elucidate the specific findings or results that led the author to draw such a conclusion. What specific data or analysis supports this assertion?

7. On line 191, it is suggested to tone down the statement: "Therefore, among the components of MetS, reducing WC is particularly important for reducing the risk of RVO," as the results of this study suggests an association and cannot establish a causal relationship.

8. On line 201, please consider referencing a study that suggests menopause may influence RVO risk, such as:

Hwang S, Kang SW, Choi KJ, Son KY, Lim DH, Shin DW, Choi D, Kim SJ. Early menopause is associated with an increased risk of retinal vascular occlusions: a nationwide cohort study. Sci Rep. 2022 Apr 12;12(1):6068. doi: 10.1038/s41598-022-10088-0. PMID: 35414644;

9. The limitations inherent in this type of research, which relies on claim data, should be thoroughly addressed. Specifically, the inability to independently verify diagnoses and the fact that the study only captures patients who visited healthcare services need to be acknowledged.

10. It is important to note that the initial study sample of 6,891,399 individuals primarily comprises those who underwent health examinations, potentially limiting its representativeness for the broader population. Additionally, the selection criteria, focusing on individuals who underwent health examinations four times over four years, may not accurately reflect 20s and 30s of general population. Furthermore, the high male representation within the study participants should also be acknowledged. Therefore, it is imperative to acknowledge these limitations within the manuscript for transparency and completeness.

REVIEWER 2

The authors examined the impact of the cumulative burden of metabolic syndrome on the incidence of retinal vein occlusion in young adults. They concluded that rapid detection and treatment of metabolic disorders in young adults is important to reduce the risk of retinal vein occlusion. As described below, there are several areas for improvement.

1. there are serious concerns about selection bias in the selection of study subjects. Populations under the age of 40 who receive annual health checks are more likely to have more systemic diseases or higher socioeconomic status, and this should be taken into account.

2. Even among populations that receive annual health checks, data may be missing for some of the parameters examined in this study, such as waist circumference. Please explain how you handled this.

3. While most other studies set the washout period at a minimum of 2 years and sometimes longer, this study set it at 1 year. Therefore, it is possible that patients with previously diagnosed RVOs were included rather than patients with new onset, and this needs further explanation.

4. line 104 states that all RVOs were included based on ICD-10, and the legend to Figure 2 states that they were categorized using KCD disease classification codes. To my knowledge, the two-digit decimal codes used for figure legends have been in use since KCD-6 in 2010. Please explain the exact classification used in this study in the Methods section.

5. The chronic kidney disease diagnosis code does not indicate a specific diagnosis, so it is better to use the eGFR measurements rather than CKD code.

6. If you used multiple models in your evaluation, please describe your results using the commonly used Charlson-Comorbiditi index.

7. The survey items were slightly different for each year, which may have resulted in inconsistent survey questions for certain items such as alcohol drinking and exercise. Please explain how you handled this.

8. The incidence of metabolic syndrome and RVO was very low in people under age 30, which may be due to a stronger association with systemic diseases such as leukemia other than metabolic syndrome. Have you done any further analysis on this?

9. You mentioned that there is a difference between men and women, but the study was based on younger people, so it is possible that the difference is due to contraceptive use. If you have analyzed differences due to contraceptive use, please present the results.

10. You cited another study in the discussion section, did this study also find differences in other parameters including WC based on DM? Please present the results.

Reviewers' comments:

Reviewer's Responses to Questions

**Comments to the Author**

1. Is the manuscript technically sound, and do the data support the conclusions?

Reviewer #1: Yes

Reviewer #2: Partly

2. Has the statistical analysis been performed appropriately and rigorously? 

Reviewer #1: Yes

Reviewer #2: Yes

3. Have the authors made all data underlying the findings in their manuscript fully available?

Reviewer #1: Yes

Reviewer #2: Yes

4. Is the manuscript presented in an intelligible fashion and written in standard English?

Reviewer #1: Yes

Reviewer #2: Yes

5. Review Comments to the Author

Reviewer #1: Kim et al. demonstrated the impact of the longitudinal cumulative burden of METS on RVO in a young population. Given the significant impact of eye diseases in the young population, this research holds considerable value. Moreover, the examination of cumulative burden is an interesting aspect of the study. However, there are several points that need to be addressed:

1. While it is acknowledged that disease burden is higher in the young population, it would be beneficial to mention how the RVO incidence in the 20-40 age group compares to the overall population incidence or the incidence in the elderly. Since there have been similar studies conducted in Korea previously, referencing these studies for comparison would enhance the discussion.

2. On line 82, Figure 2 precedes Figure 1 in the manuscript. It is advisable to reverse the order for clarity.

3. Regarding line 82, please provide a rationale for the selection of a 1-year lag period.

4. Regarding Figure 2, does this study include individuals who underwent health examinations four times between 2009 and 2012? If so, the index data should presumably be in 2012. However, in Figure 2, it mentions the exclusion of cases diagnosed with RVO from 2002 to 2009 and within one year from the index date. Does this mean that individuals diagnosed with RVO in 2010 were not excluded? I assume that individuals diagnosed with RVO before the index date might have been excluded. The flow chart requires further clarification to provide a clearer understanding.

5. On line 167, the conclusion "among the five metabolic components, increased WC had the greatest increase in RVO risks" needs further clarification. What criteria were used to reach this conclusion? Among the components, it appears that impaired fasting glucose has the highest risk when there are four times serial abnormalities. Please elaborate.

6. On line 177, the statement, "it can be inferred that the risk of RVO increases more than four times in people diagnosed with MetS during health examinations," requires additional clarification. It is crucial to elucidate the specific findings or results that led the author to draw such a conclusion. What specific data or analysis supports this assertion?

7. On line 191, it is suggested to tone down the statement: "Therefore, among the components of MetS, reducing WC is particularly important for reducing the risk of RVO," as the results of this study suggests an association and cannot establish a causal relationship.

8. On line 201, please consider referencing a study that suggests menopause may influence RVO risk, such as:

Hwang S, Kang SW, Choi KJ, Son KY, Lim DH, Shin DW, Choi D, Kim SJ. Early menopause is associated with an increased risk of retinal vascular occlusions: a nationwide cohort study. Sci Rep. 2022 Apr 12;12(1):6068. doi: 10.1038/s41598-022-10088-0. PMID: 35414644;

9. The limitations inherent in this type of research, which relies on claim data, should be thoroughly addressed. Specifically, the inability to independently verify diagnoses and the fact that the study only captures patients who visited healthcare services need to be acknowledged.

10. It is important to note that the initial study sample of 6,891,399 individuals primarily comprises those who underwent health examinations, potentially limiting its representativeness for the broader population. Additionally, the selection criteria, focusing on individuals who underwent health examinations four times over four years, may not accurately reflect 20s and 30s of general population. Furthermore, the high male representation within the study participants should also be acknowledged. Therefore, it is imperative to acknowledge these limitations within the manuscript for transparency and completeness.

These points, once addressed, will significantly enhance the quality and comprehensibility of the manuscript.

Reviewer #2: The authors examined the impact of the cumulative burden of metabolic syndrome on the incidence of retinal vein occlusion in young adults. They concluded that rapid detection and treatment of metabolic disorders in young adults is important to reduce the risk of retinal vein occlusion. As described below, there are several areas for improvement.

1. there are serious concerns about selection bias in the selection of study subjects. Populations under the age of 40 who receive annual health checks are more likely to have more systemic diseases or higher socioeconomic status, and this should be taken into account.

2. Even among populations that receive annual health checks, data may be missing for some of the parameters examined in this study, such as waist circumference. Please explain how you handled this.

3. While most other studies set the washout period at a minimum of 2 years and sometimes longer, this study set it at 1 year. Therefore, it is possible that patients with previously diagnosed RVOs were included rather than patients with new onset, and this needs further explanation.

4. line 104 states that all RVOs were included based on ICD-10, and the legend to Figure 2 states that they were categorized using KCD disease classification codes. To my knowledge, the two-digit decimal codes used for figure legends have been in use since KCD-6 in 2010. Please explain the exact classification used in this study in the Methods section.

5. The chronic kidney disease diagnosis code does not indicate a specific diagnosis, so it is better to use the eGFR measurements rather than CKD code.

6. If you used multiple models in your evaluation, please describe your results using the commonly used Charlson-Comorbiditi index.

7. The survey items were slightly different for each year, which may have resulted in inconsistent survey questions for certain items such as alcohol drinking and exercise. Please explain how you handled this.

8. The incidence of metabolic syndrome and RVO was very low in people under age 30, which may be due to a stronger association with systemic diseases such as leukemia other than metabolic syndrome. Have you done any further analysis on this?

9. You mentioned that there is a difference between men and women, but the study was based on younger people, so it is possible that the difference is due to contraceptive use. If you have analyzed differences due to contraceptive use, please present the results.

10. You cited another study in the discussion section, did this study also find differences in other parameters including WC based on DM? Please present the results.

6. PLOS authors have the option to publish the peer review history of their article (what does this mean?). If published, this will include your full peer review and any attached files.

Reviewer #1: No

Reviewer #2: No

---

## [Author Response · Author response to Decision Letter 0]

20 Jan 2024

*General comments

- Response: Thank you for the instructions and insightful comments. The manuscript has been revised in accordance with the additional comments.

*Comment 1. Please ensure that your manuscript meets PLOS ONE's style requirements, including those for file naming. The PLOS ONE style templates can be found at 

- Response: We revised the manuscript according to PLOS ONE's style requirements.

*Comment 2. In your Data Availability statement, you have not specified where the minimal data set underlying the results described in your manuscript can be found. PLOS defines a study's minimal data set as the underlying data used to reach the conclusions drawn in the manuscript and any additional data required to replicate the reported study findings in their entirety. All PLOS journals require that the minimal data set be made fully available. For more information about our data policy, please see http://journals.plos.org/plosone/s/data-availability.

- Response: The following Data Availability statement has been added in the revised manuscript (page 1, line 18).

Data Availability Statement: Access to raw data from the Korean Health Insurance Review and Assessment (HIRA) service is regulated by the Rules for Data Exploration and Use of the HIRA. Data are available from the Health Insurance Review and Assessment Service database for researchers who meet the criteria for access to confidential data after receiving approval from the HIRA Data Access Committee. The HIRA data can be obtained using the website (http://opendata.hira.or.kr). 

*Comment 3. Please include your full ethics statement in the ‘Methods’ section of your manuscript file. In your statement, please include the full name of the IRB or ethics committee who approved or waived your study, as well as whether or not you obtained informed written or verbal consent. If consent was waived for your study, please include this information in your statement as well. 

- Response: The following paragraph has been inserted into the Methods section of the revised manuscript:

(After revision, page 3, line 62)

The methods of Ahn et al. were grossly referenced in this study[12]. This study was approved by the Institutional Review Board (IRB) of Kim’s Eye Hospital (Kim’s Eye Hospital IRB; IRB number KEH 2022-04-008), and was conducted according to the principles of the Declaration of Helsinki. The need for informed consent was waived by the IRB of Kim’s Eye Hospital, because the study did not include any identifiable information about the subjects. The Korean National Health Insurance Corporation has allowed authors to use the database based on the approved IRB. The data access dates for research purposes were June 2, 2022

REVIEWER 1

*Comment. Kim et al. demonstrated the impact of the longitudinal cumulative burden of METS on RVO in a young population. Given the significant impact of eye diseases in the young population, this research holds considerable value. Moreover, the examination of cumulative burden is an interesting aspect of the study. However, there are several points that need to be addressed:

- Response: Thank you for the comments. We have made our best efforts to address the issues raised by the reviewer. We believe that the manuscript has undergone significant improvements. We hope that our responses address the concerns regarding this manuscript. Detailed responses to the reviewers’ comments are provided below.

*Comment 1. While it is acknowledged that disease burden is higher in the young population, it would be beneficial to mention how the RVO incidence in the 20-40 age group compares to the overall population incidence or the incidence in the elderly. Since there have been similar studies conducted in Korea previously, referencing these studies for comparison would enhance the discussion.

- Response: A nationwide retrospective study using data from the Korean National Health Claims Database from 2007 to 2011 revealed the RVO incidences per 100,000 person-years. For ages 20-29, 30-39, and 40-49, the rates were 4.03, 10.32, and 32.22, respectively. The rates dramatically increased with age, with an incidence of 80.22 for age 50-59 and 172.90 for ages 60-69. This pattern shows a more than twofold increase in incidence every decade from the 20s to the 70s.

The following paragraph has been added to the revised manuscript.

(After revision, page 20, line 193)

A nationwide retrospective study using data from the Korean National Health Claims Database from 2007 to 2011 revealed the RVO incidences per 100,000 person-years.[21] For ages 20-29, 30-39, and 40-49, the rates were 4.03, 10.32, and 32.22, respectively. The rates dramatically increased with age, with incidences of 80.22 for ages 50-59 and 172.90 for ages 60-69. This pattern shows a more than twofold increase in incidence every decade from the 20s to the 70s.

*Comment 2. On line 82, Figure 2 precedes Figure 1 in the manuscript. It is advisable to reverse the order for clarity.

- Response: The positions of Figure 1 and Figure 2 were switched.

*Comment 3. Regarding line 82, please provide a rationale for the selection of a 1-year lag period.

- Response: A one-year lag period was implemented to enhance the causality of the research. This was done to mitigate the possibility of delayed RVO diagnoses due to delayed eye examinations and to exclude individuals who received regular checkups over a four-year period and passed away within one year thereafter. By excluding these possibilities, this study aimed to increase the causality of the research, measure exposure over a four-year period, and observe the occurrence of RVO one year after the measurements were taken.

The following revision has been made to clarify this point.

(After revision, page 4, line 83)

A one-year lag period was implemented to enhance the causality of the study. This was done to mitigate the possibility of delayed RVO diagnoses due to delayed eye examinations and to exclude individuals who received regular checkups over a four-year period and passed away within one year thereafter. By excluding these possibilities, this study aimed to increase the causality of the research, measure exposure over a four-year period, and observe the occurrence of RVO one year after the measurements were taken.

*Comment 4. Regarding Figure 2, does this study include individuals who underwent health examinations four times between 2009 and 2012? If so, the index data should presumably be in 2012. However, in Figure 2, it mentions the exclusion of cases diagnosed with RVO from 2002 to 2009 and within one year from the index date. Does this mean that individuals diagnosed with RVO in 2010 were not excluded? I assume that individuals diagnosed with RVO before the index date might have been excluded. The flow chart requires further clarification to provide a clearer understanding.

- Response: In the Figure 2 legend, the specific meaning of the sentence “Patients who received a health examination between January 1, 2009, and December 31, 2012 and have subsequently undergone four consecutive examinations thereafter” is as follows: For the 2009 group, participants who underwent annual health examinations in four consecutive years from 2009 to 2012; in the 2010 group, participants who underwent annual health examinations in four consecutive years from 2010 to 2013; in the 2011 group, participants who underwent annual health examinations in four consecutive years from 2011 to 2014; and in the 2012 group, participants who underwent annual health examinations in four consecutive years from 2012 to 2014. The exclusion criteria were as follows: in 2009, for example, individuals who underwent annual health examinations from 2009 to 2012, and experienced RVO or died in 2013 were excluded.

These points have been clarified. In addition, in response to comment #2, the order of Figure 1 and Figure 2 was reversed.

(After revision, page 5, figure legend for figure 1)

(For example, for 2009, participants underwent annual health examinations continuously for four years from 2009 to 2012)

(After revision, page 5, figure legend for figure 1)

(For example, for 2009, individuals who received annual health examinations from 2009 to 2012 and experienced RVO or died in 2013 were excluded.)

*Comment 5. On line 167, the conclusion "among the five metabolic components, increased WC had the greatest increase in RVO risks" needs further clarification. What criteria were used to reach this conclusion? Among the components, it appears that impaired fasting glucose has the highest risk when there are four times serial abnormalities. Please elaborate.

- Response: As you have pointed out, impaired fasting glucose has the highest risk when there are four serial abnormalities (Table 2). We revised this sentence as follows:

(In the original manuscript, discussion section)

(3) among the five metabolic components, increased WC had the greatest increase in RVO risks

Among the five diagnostic components of MetS, the impact of WC on RVO occurrence was particularly significant.

(After revision, page 20, line 206)

(3) among the five metabolic components, impaired fasting glucose showed the greatest increase in RVO risk.

(After revision, page 21, line 219)

Among the five diagnostic components of MetS, the impact of WC on RVO occurrence was particularly significant when individuals with impaired fasting glucose, prediabetes, or diabetes were excluded.

*Comment 6. On line 177, the statement, "it can be inferred that the risk of RVO increases more than four times in people diagnosed with MetS during health examinations," requires additional clarification. It is crucial to elucidate the specific findings or results that led the author to draw such a conclusion. What specific data or analysis supports this assertion?

- Response: This sentence appears to have been misrepresented during the English language correction process. We meant to refer to four diagnoses, not to a risk four times as high. We have revised this to the following sentence: 

(After revision, page 20, line 216)

"Given the proportional increase in RVO risk according to the degree of metabolic burden with temporal changes, it is plausible that the risk of RVO would be higher in patients diagnosed with MetS more than four times during their health checkups."

*Comment 7. On line 191, it is suggested to tone down the statement: "Therefore, among the components of MetS, reducing WC is particularly important for reducing the risk of RVO," as the results of this study suggests an association and cannot establish a causal relationship.

- Response: We agree with your comment that a causal relationship cannot be proven based on the study results. The following revisions have been made to improve the clarity of the statement.

(In the original manuscript, discussion section)

Therefore, among the components of MetS, reducing WC is particularly important for reducing the risk of RVO.

(After revision, page 21, line 232)

Further studies are required to determine whether there is a definitive causal relationship between WC and MetS.

*Comment 8. On line 201, please consider referencing a study that suggests menopause may influence RVO risk, such as:

Hwang S, Kang SW, Choi KJ, Son KY, Lim DH, Shin DW, Choi D, Kim SJ. Early menopause is associated with an increased risk of retinal vascular occlusions: a nationwide cohort study. Sci Rep. 2022 Apr 12;12(1):6068. doi: 10.1038/s41598-022-10088-0. PMID: 35414644;

- Response: The following revisions have been made to include the potential effect of menopause on the risk of RVO.

(After revision, page 21, line 243)

Furthermore, a recent study showed that early menopause is associated with an increased risk of RVO.[36]

(After revision, page 27, line 371)

A new reference has been added.

*Comment 9. The limitations inherent in this type of research, which relies on claim data, should be thoroughly addressed. Specifically, the inability to independently verify diagnoses and the fact that the study only captures patients who visited healthcare services need to be acknowledged.

- Response: Thank you for the comment. We have added the following paragraph to the Limitations section: 

(After revision, page 22, lines 252)

Our study had several limitations. First, using a claims database focused solely on healthcare utilization to identify patients with RVO may have underrepresented asymptomatic cases. Therefore, our data reflected the incidence of clinically diagnosed RVO, potentially underestimating the true occurrence of RVO. Second, the lack of access to hospital-based medical records to validate RVO occurrences and review clinical data compromised the accuracy of this study's data. These limitations include the risk of misclassifying diagnoses and the inability to assess factors such as severity, extent, prognosis, or patient comorbidities. Third, owing to the inherent constraints of studies utilizing claims databases, clinical characteristics and interobserver variability in RVO diagnosis were not standardized.

*Comment 10. It is important to note that the initial study sample of 6,891,399 individuals primarily comprises those who underwent health examinations, potentially limiting its representativeness for the broader population. Additionally, the selection criteria, focusing on individuals who underwent health examinations four times over four years, may not accurately reflect 20s and 30s of general population. Furthermore, the high male representation within the study participants should also be acknowledged. Therefore, it is imperative to acknowledge these limitations within the manuscript for transparency and completeness.

- Response: The following sentences have been added to the limitations section: 

(After revision, page 22, line 260)

Fourth, male patients comprised over two-thirds of the participants, owing to the higher probability of employees undergoing routine health checkups at their workplaces. This sex imbalance introduces a potential selection bias, limiting the representation of both sexes in a broader population. Furthermore, the selection criteria, emphasizing individuals who underwent health assessments four times over four years, might not fully capture the demographics of individuals in their 20s and 30s within the general population.

REVIEWER 2

*Comment. The authors examined the impact of the cumulative burden of metabolic syndrome on the incidence of retinal vein occlusion in young adults. They concluded that rapid detection and treatment of metabolic disorders in young adults is important to reduce the risk of retinal vein occlusion. As described below, there are several areas for improvement.

*Comment 1. there are serious concerns about selection bias in the selection of study subjects. Populations under the age of 40 who receive annual health checks are more likely to have more systemic diseases or higher socioeconom

---

## [Decision Letter · Decision Letter 1]

20 Mar 2024

PONE-D-23-20018R1Cumulative effect of metabolic syndrome on the risk of retinal vein occlusion in young patients: a nationwide population-based studyPLOS ONE

Dear Dr. Kim,

Thank you for submitting your manuscript to PLOS ONE. After careful consideration, we feel that it has merit but does not fully meet PLOS ONE’s publication criteria as it currently stands. Therefore, we invite you to submit a revised version of the manuscript that addresses the points raised during the review process.

We look forward to receiving your revised manuscript.

Kind regards,

Jiro Kogo

Academic Editor

PLOS ONE

Journal Requirements:

Reviewers' comments:

Reviewer's Responses to Questions

**Comments to the Author**

1. If the authors have adequately addressed your comments raised in a previous round of review and you feel that this manuscript is now acceptable for publication, you may indicate that here to bypass the “Comments to the Author” section, enter your conflict of interest statement in the “Confidential to Editor” section, and submit your "Accept" recommendation.

Reviewer #1: All comments have been addressed

Reviewer #2: All comments have been addressed

2. Is the manuscript technically sound, and do the data support the conclusions?

Reviewer #1: Yes

Reviewer #2: Yes

3. Has the statistical analysis been performed appropriately and rigorously? 

Reviewer #1: Yes

Reviewer #2: Yes

4. Have the authors made all data underlying the findings in their manuscript fully available?

Reviewer #1: Yes

Reviewer #2: Yes

5. Is the manuscript presented in an intelligible fashion and written in standard English?

Reviewer #1: Yes

Reviewer #2: Yes

6. Review Comments to the Author

Reviewer #1: All comments have been addressed.

One more comment to add. Please double-check and amend the discrepancy between the methodology section, which states the use of the National Health Insurance Service (NHIS)'s Database, and the acknowledgment and data availability statement, which indicate the use of Health Insurance Review and Assessment Service (HIRA) data.

Reviewer #2: The authors examined the impact of the cumulative burden of metabolic syndrome on the incidence of retinal vein occlusion in young adults. They concluded that rapid detection and treatment of metabolic disorders in young adults is important to reduce the risk of retinal vein occlusion.

All comments has been addressed well.

7. PLOS authors have the option to publish the peer review history of their article (what does this mean?). If published, this will include your full peer review and any attached files.

Reviewer #1: No

Reviewer #2: No

---

## [Author Response · Author response to Decision Letter 1]

4 Apr 2024

*General comments

- Response: Thank you for the instructions and comments. I have double-checked the references on the manuscript, and there were no retracted papers found. 

Reviewer #1

Please double-check and amend the discrepancy between the methodology section, which states the use of the National Health Insurance Service (NHIS)'s Database, and the acknowledgment and data availability statement, which indicate the use of Health Insurance Review and Assessment Service (HIRA) data.

- Response: Thank you for the comment. I have checked the discrepancy between the methods section and the data availability statement and acknowledgment, and revised the data availability statement and acknowledgment as follows. 

(After revision, page 1, line 18)

Data Availability Statement: Data are available from the Korea National Health Insurance Sharing Service (https://nhiss.nhis.or.kr/bd/ay/bdaya001iv.do;jsessionid=KiOt6ilaTGXMmR1mbeeA9Ax7nJjIYDsKr4UJAJvHhNyo511E0PovM1WfrGSaegL6.primrose22_servlet_engine1, Tel.: (82) 33-736-2432, 2433) for researchers who meet the criteria for access to confidential data.

(After revision, page 24, line 279)

Acknowledgments

The authors wish to thank the National Health Insurance Service for providing the data regarding the original cohort.

---

## [Editor Report · Decision Letter 2]

2 May 2024

Cumulative effect of metabolic syndrome on the risk of retinal vein occlusion in young patients: a nationwide population-based study

PONE-D-23-20018R2

Dear Dr.KIm

We’re pleased to inform you that your manuscript has been judged scientifically suitable for publication and will be formally accepted for publication once it meets all outstanding technical requirements.

Kind regards,

Jiro Kogo

Academic Editor

PLOS ONE
---

## [Editor Report · Acceptance letter]

7 May 2024

PONE-D-23-20018R2 

PLOS ONE

Dear Dr. Kim, 

I'm pleased to inform you that your manuscript has been deemed suitable for publication in PLOS ONE. Congratulations! Your manuscript is now being handed over to our production team.

Kind regards, 

on behalf of

Prof. Jiro Kogo 

Academic Editor

PLOS ONE